# Learning Independent Features with Adversarial Nets for Non-linear ICA

## Abstract

Reliable measures of statistical dependence could be useful tools for learning independent features and performing tasks like source separation using Independent Component Analysis (ICA). Unfortunately, many of such measures, like the mutual information, are hard to estimate and optimize directly. We propose to learn independent features with adversarial objectives (Goodfellow et al., 2014; Arjovsky et al., 2017; Huszar, 2016) which optimize such measures implicitly. These objectives compare samples from the joint distribution and the product of the marginals without the need to compute any probability densities. We also propose two methods for obtaining samples from the product of the marginals using either a simple resampling trick or a separate parametric distribution. Our experiments show that this strategy can easily be applied to different types of model architectures and solve both linear and non-linear ICA problems.

## 1 Introduction

When unsupervised models are developed to learn *interesting* features of data, this often implies that each feature represents some interesting unique property of the data that cannot easily be derived from the other features. A sensible way to learn such features is by ensuring that they are *statistically independent*. However, the removal of all possible dependencies from a set of features is not an easy task. Therefore, research on the extraction of such statistically independent features has so far mostly focussed on the linear case, which is known as Independent Component Analysis (ICA; Hyvärinen et al. 2004). Linear ICA has many applications and has been especially popular as a method for blind source separation (BSS) and its application domains include medical signal analysis (e.g., EEG and ECG), audio source separation and image processing (see Naik & Kumar 2011 for a comprehensive overview of ICA applications). Unfortunately, the linear ICA model is not always appropriate and in the light of the recent success of deep learning methods, it would be interesting to learn more general non-linear feature transformations. Some non-linear and also non-iid ICA methods have been proposed (Almeida, 2003; Hyvarinen & Morioka, 2017) but the learning of general non-linear ICA models is both ill-defined and still far from being solved.

One obstacle in way of learning independent features is the lack of measures of independence that are easy to optimize. Many learning algorithms for linear and non-linear Independent Component Analysis are in some way based on a minimization of the mutual information (MI) or similar measures which compare a joint distribution with the product of its marginals. Such measures are typically hard to estimate in practice. This is why a popular ICA method like Infomax (Bell & Sejnowski, 1995), for example, only minimizes the mutual information indirectly by maximizing the joint entropy of the features instead. While there is some work on estimators for mutual information and independence based non-parametric methods (Kraskov et al., 2004; Gretton et al., 2005), it is typically not straightforward to employ such measures as optimization criteria.

Recently, the framework of Generative Adversarial Networks (GANs) was proposed for learning generative models (Goodfellow et al., 2014) and matching distributions. GAN training can be seen as approximate minimization of the Jensen-Shannon divergence between two distributions without the need to compute densities. Other recent work extended this interpretation of GANs to other divergences and distances between distributions (Arjovsky et al., 2017; Hjelm et al., 2017; Mao et al., 2016). While most work on GANs applies this matching of distributions in the context of genera-

tive modelling, some recent work has extended these ideas to learn features which are invariant to different domains (Ganin et al., 2016) or noise conditions (Serdyuk et al., 2016).

We show how the GAN framework allows us to define new objectives for learning statistically independent features. The gist of the idea is to use adversarial training to train some joint distribution to produce samples which become indistinguishable from samples of the product of its marginals. Our empirical work shows that auto-encoder type models which are trained to optimize our independence objectives can solve both linear and non-linear ICA problems with different numbers of sources and observations. Unlike many other ICA methods, our objective is more or less independent of the mixing function and doesn't use properties that are specific for the linear or so-called 'post-nonlinear' case. We also propose a heuristic for model selection for such architectures that seems to work reasonably well in practice.

## 2 BACKGROUND

The mutual information of two stochastic variables $Z_1$ and $Z_2$ corresponds to the Kullback-Leibler (KL) divergence between their joint density and the product of the marginal densities:

$$I(Z_1, Z_2) = \int \int p(z_1, z_2) \log \frac{p(z_1, z_2)}{p(z_1)p(z_2)} \mathrm{d}z_1 z_2. \tag{1}$$

We will often write densities like $p(Z_1 = z_1)$ as $p(z_1)$ to save space. The MI is zero if and only if all the variables are mutually independent. One benefit of working with MI as a measure of dependence/independence, is that it can easily be related to other information theoretical quantities, like for example the entropies of the distributions involved. Another nice property of the mutual information is that, unlike differential entropy, it is invariant under reparametrizations of the marginal variables (Kraskov et al., 2004). This means that if two functions $f$ and $g$ are homeomorphisms, the $I(Z_1, Z_2) = I(f(Z_1), g(Z_2))$. Unfortunately, the mutual information is often hard to compute or estimate, especially for high dimensional sample spaces.

Generative Adversarial Networks (GANs; Goodfellow et al. 2014) provide a framework for matching distributions without the need to compute densities. During training, two neural networks are involved: the *generator* and the *discriminator*. The generator is a function $G(\cdot)$ which maps samples from a known distribution (e.g., the unit variance multivariate normal distribution) to points that live in the same space as the samples of the data set. The discriminator is a classifier $D(\cdot)$ which is trained to separate the 'fake' samples from the generator from the 'real' samples in the data set. The parameters of the generator are optimized to 'fool' the discriminator and maximize its loss function using gradient information propagated through the samples.[1] In the original formulation of the GAN framework, the discriminator is optimized using the cross-entropy loss. The full definition of the GAN learning objective is given by

$$\min_G \max_D E_{\text{data}}[\log D(\mathbf{x})] + E_{\text{gen}}[\log(1 - D(\mathbf{y}))], \tag{2}$$

where $E_{\text{data}}$ and $E_{\text{gen}}$ denote expectations with respect to the data and generator distributions.

Since we will evaluate our models on ICA source separation problems, we will describe this setting in a bit more detail as well. The original linear ICA model assumes that some observed multivariate signal $\mathbf{x}$ can be modelled as

$$\mathbf{x} = \mathbf{A}\mathbf{s}, \tag{3}$$

where $\mathbf{A}$ is a linear transformation and $\mathbf{s}$ is a set of mutually independent *source* signals which all have non-Gaussian distributions. Given the observations $\mathbf{x}$, the goal is to retrieve the source signals $\mathbf{s}$. When the source signals are indeed non-Gaussian (or there is at least no more than one Gaussian source), the matrix $\mathbf{A}$ is of full column rank and the number of observations is at least as large as the number of sources, linear ICA is guaranteed to be identifiable (Comon, 1994) up to a permutation and rescaling of the sources. When the mixing of the signals is not linear, identifiability cannot be guaranteed in general and the problem is ill-posed. However, under certain circumstances, some specific types of non-linear mixtures like post non-linear mixtures (PNL) can still be separated (Taleb & Jutten, 1999). Separability can sometimes also be observed (albeit not guaranteed) when the number of mixtures/observations is larger than the number of source signals.

---

[1] While this restricts the use of GANs to continuous distributions, methods for discrete distributions have been proposed as well (Hjelm et al., 2017).

## 3 MINIMIZING AND MEASURING DEPENDENCE

For the moment, assume that we have access to samples from both the joint distribution $p(\mathbf{z})$ and the product of the marginals $\prod_i p(z_i)$. We now want to measure how dependent/independent the individual variables of the joint distribution are without measuring any densities. As pointed out by Arjovsky et al. (2017), the earth mover's distance between two distributions $q$ and $r$ can, under certain conditions, be approximated by letting $f(\cdot)$ be a neural network and solving the following optimization problem:

$$\max_{\|f\|_L \leq 1} E_{\mathbf{z} \sim q(\mathbf{z})}[f(\mathbf{z})] - E_{\mathbf{z} \sim r(\mathbf{z})}[f(\mathbf{z})]. \tag{4}$$

If we substitute $q$ and $r$ for $p(\mathbf{z})$ and $\prod_i p(z_i)$, respectively, we can consider Equation 4 to be an approximate measure of dependence for $p(\mathbf{z})$ that can serve as an alternative to the mutual information in the sense that the objective it approximates will be zero if and only if the distributions are the same and therefore the variables are independent. Now if it is also possible to backpropagate gradients through the samples with respect to the parameters of the distribution, we can use these to minimize Equation 4 and make the variables more independent. Similarly, the standard GAN objective can be used to approximately minimize the JS-divergence between the joint and marginal distributions instead. While we focussed on learning independent features and the measuring of dependence is not the subject of the research in this paper, we think that the adversarial networks framework may provide useful tools for this as well.

Finally, as shown in a blog post (Huszar, 2016), the standard GAN objective can also be adapted to approximately optimize the KL-divergence. This objective is obviously an interesting case because it results in an approximate optimization of the mutual information itself but in preliminary experiments we found it harder to optimize than the more conventional GAN objectives.

### 3.1 OBTAINING THE SAMPLES

So far, we assumed that we had access to both samples from the joint distribution and from the product of the marginals. To obtain approximate samples from the product of the marginals, we propose to either *resample* the values of samples from the joint distribution or to train a separate generator network with certain architectural constraints.

Given a sample $(z_1, \ldots, z_M)^{\mathsf{T}}$ of some joint distribution $p(z_1, \ldots, z_M)$, a sample of the marginal distribution $p(z_1)$ can be obtained by simply discarding all the other variables from the joint sample. To obtain samples from the complete product $\prod_{i=1}^{M} p(z_i)$, the same method can be used by taking $M$ samples from the joint distribution and making sure that each of the $M$ dimensions from the new factorized sample is taken from a different sample of the joint. In other words, given $K$ joint samples where $K \geq M$, one can randomly choose $M$ integers from $\{1, \ldots, N\}$ without replacement and use them to select the elements of the sample from the factorized distribution. When using sampling *with* replacement, a second sample obtained in this way from the same batch of joint samples would not be truly independent of the first. We argue that this is not a big problem in practice as long as one ensures that the batches are large enough and randomly chosen.

Another way to simulate the product of marginal distributions is by using a separate generator network which is trained to optimize the same objective as the generator of the joint distribution. By sampling independent latent variables and transforming each of them with a separate multi-layer perceptron, this generator should be able to learn to approximate the joint distribution with a factorized distribution without imposing a specific prior. While it may be more difficult to learn the marginal distributions explicitly, it could in some situations be useful to have an explicit model for this distribution available after training for further analysis or if the goal is to build a generative model. While the resampling method above is obviously simpler, this parameterized approach may be especially useful when the data are not iid (like in time series) and one doesn't want to ignore the inter-sample dependencies.

## 4 ADVERSARIAL NON-LINEAR INDEPENDENT COMPONENTS ANALYSIS

As a practical application of the ideas described above, we will now develop a system for learning independent components. The goal of the system is to learn an *encoder* network $F(\mathbf{x})$ which maps

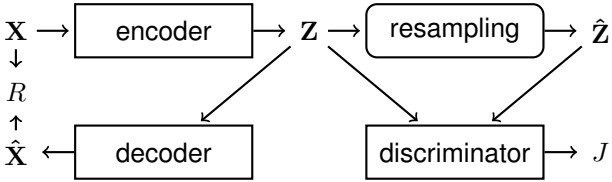

Figure 1: Schematic representation of the entire system for learning non-linear ICA. Specific functional shapes can be enforced by choosing a suitable decoder architecture.

data/signals to informative features $\mathbf{z}$ which are mutually independent. We will use an adversarial objective to achieve this in the manner described above. However, enforcing independence by itself does not guarantee that the mapping from the observed signals $\mathbf{x}$ to the predicted sources $\mathbf{z}$ is informative about the input. To enforce this, we add a *decoder* network $V(\mathbf{z})$, which tries to reconstruct the data from the predicted features as was done by Schmidhuber (1992).

Given the encoder, the decoder, the discriminator, samples from the data, the joint distribution, and the product of the marginals, we can now compute the GAN objective from Equation 2 (or Equation 4, assuming the Lipschitz constraint is also enforced) and add the reconstruction objective to it. This leads to the following total objective function:

$$O = E_{\mathbf{x} \sim p(\mathbf{x})} \left[ J(\mathbf{x}) + \lambda \|\mathbf{x} - V(F(\mathbf{x}))\|_{2,1} \right], \tag{5}$$

where $\| \cdot \|_{2,1}$ is the $L_1$ norm and $\lambda$ is a hyperparameter which controls the trade-off between the reconstruction quality and the adversarial independence criterion $J$. Figure 1 shows a schematic representation of the training setup in its entirety. The full procedure of our setup using the standard GAN objective is given by Algorithm 1.

---

**Algorithm 1** Adversarial Non-linear ICA train loop

---

**input** data $\mathcal{X}$, encoder $F$, decoder $V$, discriminator $D$, (optional) generator $G$
  **while** Not converged **do**
    sample a batch $\mathbf{X}$ of $N$ data (column) vectors from $\mathcal{X}$
    $\mathbf{Z} \leftarrow F(\mathbf{X})$ // apply encoder
    $\hat{\mathbf{X}} \leftarrow V(\mathbf{Z})$ // apply decoder
    $\hat{\mathbf{Z}} \leftarrow \text{Resample}(\mathbf{Z})$ or sample $\hat{\mathbf{Z}} \sim G(\hat{\mathbf{Z}})$
    $J = \log(D(\mathbf{Z})) + \log(1 - D(\hat{\mathbf{Z}}))$
    Update $D$ to maximize $J$
    $R = \|\mathbf{X} - \hat{\mathbf{X}}\|_{2,1}$
    Update $F$, $V$ and possibly $G$ to minimize $J + \lambda R$
  **end while**

---

Finally, we found that it is important to normalize the features vectors within a batch before permuting them and presenting them to the discriminator. This prevents them both from going to zero and from growing indefinitely in magnitude, potentially causing the discriminator to fail because it cannot keep up with the overall changes of the feature distribution. We also used these normalized features as input for the decoder, followed by an element-wise rescaling using trainable parameters, similar to what is done in batch normalization (Ioffe & Szegedy, 2015). Without normalization of the decoder inputs, the models would sometimes get stuck in degenerate solutions.

## 5 RELATED WORK

Many optimization methods for ICA are either based on non-Gaussianity, like the popular FastICA algorithm (Hyvärinen & Oja, 1997), or on minimization of the mutual information of the extracted source signals, as is implicitly done with Infomax methods (Bell & Sejnowski, 1995). The Infomax ICA algorithm maximizes the joint entropy of the estimated signals. Given a carefully constructed architecture, the marginal entropies are bounded and the maximization leads to a minimization of the mutual information. Infomax has been extended to non-linear neural network models and the

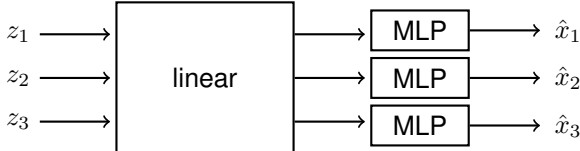

Figure 2: The decoder architecture used for the PNL experiments. It can only learn transformations in which a linear transformation is followed by the application of non-linear scalar functions to each of the dimensions.

MISEP model (Almeida, 2003) is a successful example of this. Infomax methods don't need an additional decoder component to ensure invertibility and there are no sampling methods involved. Unlike our model however, training involves a computation of the gradient of the logarithm of the determinant of the jacobian for each data point. This can be computationally demanding and also requires the number of sources and mixtures to be equal. Furthermore, our method provides a way of promoting independence of features decoupled from maximizing their information.

This work was partially inspired by Jürgen Schmidhuber's work on the learning of binary factorial codes (Schmidhuber, 1992). In that work, an auto-encoder is also combined with an adversarial objective, but one based on the mutual predictability of the variables rather than separability from the product of the marginals. To our knowledge, this method for learning binary codes has not yet been adapted for continuous variables. Our method has the advantage that only a single discriminator-type network is needed for any number of source signals. That said, Schmidhuber's setup doesn't require any sampling and a continuous version of it would be an interesting subject for future research.

The architectures in our experiments are also similar to Adversarial Auto-Encoders (AAEs) (Makhzani et al., 2015). In AAEs, the GAN principle is used to match the distribution at the output of an encoder when fed by the data with some prior as part of a Variational Autoencoder (VAE) (Kingma & Welling, 2013) setup. Similar to in our work, the KL-divergence between two distributions is replaced with the GAN objective. When a factorized prior is used (as is typically done), the AAE also learns to produce independent features. However, the chosen prior also forces the learned features to adhere to its specific shape and this may be in competition with the independence property. We actually implemented uniform and normal priors for our model but were not able to learn signal separation with those. That said, the recently proposed $\beta$-VAE (Higgins et al., 2016) has been analysed in the context of learning disentangled representations and appears to be quite effective at doing so. Another recent related model is InfoGAN (Chen et al., 2016). InfoGAN is a generative GAN model in which the mutual information between some latent variables and the outputs is maximized. While this also promotes independence of some of the latent variables, the desired goal is now to provide more control over the generated samples.

Some of the more successful estimators of mutual information are based on nearest neighbor methods which compare the relative distances of complete vectors and individual variables (Kraskov et al., 2004). An estimator of this type has also been used to perform linear blind source separation using an algorithm in which different rotations of components are compared with each other (Stögbauer et al., 2004). Unfortunately, this estimator is biased when variables are far from independent and not differentiable, limiting it's use as a general optimization criterion. Other estimators of mutual information and dependence/independence in general are based on kernel methods (Gretton et al., 2005; 2008). These methods have been very successful at linear ICA but have, to our knowledge, not been evaluated on more general non-linear problems.

Finally, there has been recent work on invertible non-linear mappings that allows the training of tractable neural network latent variable models which can be interpreted as non-linear independent component analysis. Examples of these are the NICE and real-NVP models (Dinh et al., 2014; 2016). An important difference with our work is that these models require one to specify the distribution of the source signals in advance.

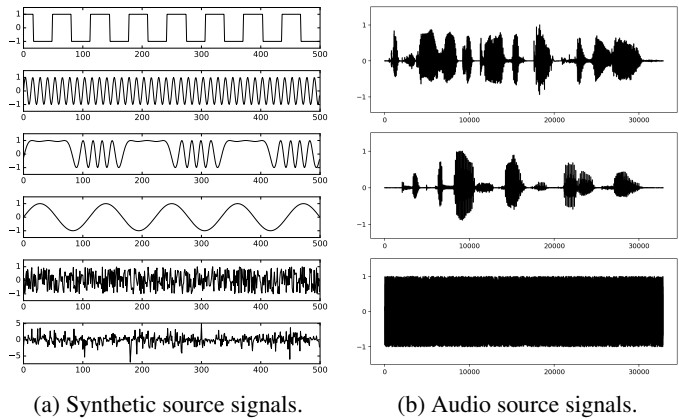

(a) Synthetic source signals.  (b) Audio source signals.

Figure 3: Source signals used in the experiments.

## 6 EXPERIMENTS

We looked at linear mixtures, post non-linear mixtures which are not linear but still separable and overdetermined general non-linear mixtures which may not be separable. ICA extracts the source signals only up to a permutation and scaling. Therefore, all results are measured by considering all possible pairings of the predicted signals and the source signals and measuring the average absolute correlation of the best pairing. We will just refer to this as $\rho_{\max}$ or simply 'correlation'. We will refer to our models with the name 'Anica', which is short for Adversarial Non-linear Independent Component Analysis.

**Source signals**  We used both synthetic signals and actual speech signals as sources for our experiments (see Figure 3). The synthetic source signals were created with the goal to include both sub-gaussian and super-gaussian distributions, together with some periodic signals for visualization purposes. The data set consisted of the first 4000 samples of these signals.[2] For the audio separation tasks, we used speech recordings from the 16kHz version of the freely available TSP data set (Kabal, 2002). The first source was an utterance from a male speaker (`MA02_04.wav`), the second source an utterance from a female speaker (`FA01_03.wav`), and the third source was uniform noise. All signals were normalized to have a peak amplitude of 1. The signals were about two seconds long, which translated to roughly 32k samples.

**Linear ICA**  As a first proof of concept, we trained a model in which both the model and the transformation of the source signals are linear. The mixed signals $\mathbf{x}$ were created by forming a matrix $\mathbf{A}$ with elements sampled uniformly from the interval $[-.5, .5]$ and multiplying it with the source signals $\mathbf{s}$. Both the encoder and decoder parts of the model were linear transformations. The discriminator network was a multilayer perceptron with one hidden layer of 64 rectified linear units.

**Post non-linear mixtures**  To generate post non-linear mixtures, we used the same procedure as we used for generating the linear mixture, but followed by a non-linear function. For the synthetic signals we used the hyperbolic tangent function. For the audio data we used a different function for each of the three mixtures: $g_1(x) = \tanh(x)$, $g_2(x) = (x + x^3)/2$ and $g_3(x) = e^x$. We found during preliminary experiments that we obtained the best results when the encoder, which learns the inverse of the mixing transformation, is as flexible as possible, while the decoder is constrained in the types of functions it can learn. One could also choose a flexible decoder while keeping the encoder constrained but this didn't seem to work well in practice. The encoder was a multi-layer perceptron (MLP) with two hidden layers of rectified linear units (ReLU; Nair & Hinton 2010). The first layer of the decoder was a linear transformation. Subsequently, each output was processed by a separate small MLP with two layers of 16 hidden ReLU units and a single input and output. This decoder was chosen to constrain the model to PNL compatible functions. Note that we did not use

---

[2]See the appendix for more details about the synthetic signals.

any sigmoid functions in our model. The discriminator network was again multilayer perceptron with one hidden layer of 64 rectified linear units.

**Over-determined multi-layer non-linear mixture**    With this task, we illustrate the benefit of our method when there are more mixture signals than sources for general non-linear mixture problem. The transformation of the source signals was $\mathbf{x} = \tanh(\cdot\mathbf{B}\tanh(\mathbf{As}))$, where $\mathbf{A}$ and $\mathbf{B}$ were randomly sampled matrices of $24 \times 6$ and $24 \times 24$ dimensions, respectively. Both the encoder and decoder for this task were MLPs with two hidden layers of ReLU units. The discriminator had two hidden layer with the same number of hidden units as was chosen for the encoder and decoder networks. There is no guarantee of identifiability for this task, but the large number of observations makes it more likely.

**Baselines**    For the linear problems, we compared our results with the FastICA (Hyvärinen & Oja, 1997) implementation from Scikit-learn (Pedregosa et al., 2011) (we report the PNL and MLP results as well just because it's possible). For the PNL problems, we implemented a version of the MISEP model (Almeida, 2003) with a neural network architecture specifically proposed for these types of problems (Zheng et al., 2007). We also computed $\rho_{\max}$ for the mixed signals. Unfortunately, we couldn't find a proper baseline for the over-determined MLP problem.

## 6.1    Optimization and hyper-parameter tuning selection

Model comparison with adversarial networks is still an open problem. We found that when we measured the sum of the adversarial loss and the reconstruction loss on held-out data, the model with the lowest loss was typically not a good model in terms of signal separation. This can for example happen when the discriminator diverges and the adversarial loss becomes very low even though the features are not independent. When one knows how the source signals are supposed to look (or sound) this may be less of a problem but even then, this would not be a feasible way to compare numerous models with different hyper-parameter settings. We found that the reliability of the score, measured as the standard deviation over multiple experiments with identical hyper-parameters, turned out to be a much better indicator of signal separation performance.

For each model, we performed a random search over the number of hidden units in the MLPs, the learning rate and the scaling of the initial weight matrices of the separate modules of the model. For each choice of hyper-parameters, we ran five experiments with different seeds. After discarding diverged models, we selected the models with the lowest standard deviation in optimization loss on a held-out set of 500 samples. We report both the average correlation scores of the model settings selected in this fashion and the ones which were highest on average in terms of the correlation scores themselves. The latter represent potential gains in performance if in future work more principled methods for GAN model selection are developed. To make our baseline as strong as possible, we performed a similar hyper-parameter search for the PNLMISEP model to select the number of hidden units, initial weight scaling and learning rate. All models were trained for 500000 iterations on batches of 64 samples using RMSProp (Tieleman & Hinton, 2012).

The standard JS-divergence optimizing GAN loss was used for all the hyper-parameter tuning experiment. We didn't find it necessary to use the commonly used modification of this loss for preventing the discriminator from saturating. We hypothesize that this is because the distributions are very similar during early training, unlike the more conventional GAN problem where one starts by comparing data samples to noise. For investigating the convergence behavior we also looked at the results of a model trained with the Wasserstein GAN loss and gradient penalty (Gulrajani et al., 2017).

## 6.2    Results

As Table 1 shows, the linear problems get solved up to very high precision for the synthetic tasks by all models, verifying that adversarial learning can be used to learn ICA. The PNL correlations obtained by the Anica models for the synthetic signals were slightly worse than of the PNLMISEP baseline. Unfortunately, the model selection procedure also didn't identify good settings for the Anica-g model and there is a large discrepancy between the chosen hyper-parameter settings and the ones that lead to the best correlation scores. The MLP results on the MLP task were high in general and the scores of the best performing hyper-parameter settings are on par with those for the PNL

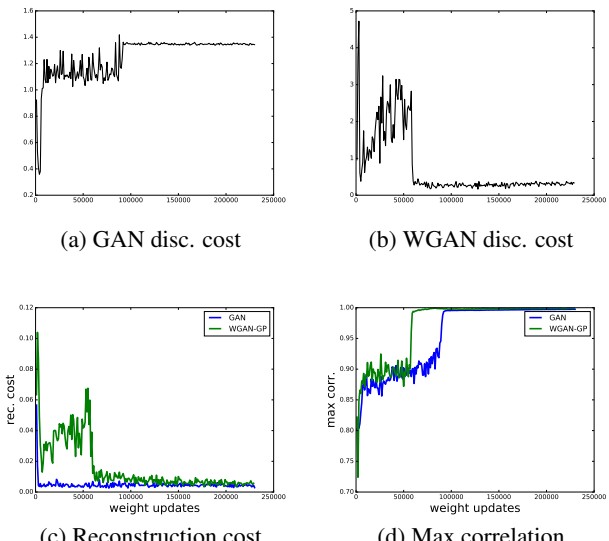

(a) GAN disc. cost   (b) WGAN disc. cost

(c) Reconstruction cost   (d) Max correlation

Figure 4: Convergence plots for the linear synthetic source task.

Table 1: Maximum correlation results on all tasks for the synthetic data. A 'g' in the suffix of the model name indicates that a separate generator network was used instead of the resampling method. Two scores separated by a '/' indicate that the first score was obtained using the model selection described in Section 6.1 while the second score is simply the best score *a posteriori*. Parentheses refer contain the standard deviation of the scores multiplied with $10^{-4}$.

| Method | Linear | PNL | MLP |
|---|---|---|---|
| Anica | .9987(6.5)/.9994(1.4) | .9794(53)/.9877(7.9) | .9667(325)/.9831(16) |
| Anica-g | .9996(1.2)/.9996(1.2) | .7098(724)/.9802(47) | .9770(33)/.9856(10.8) |
| PNLMISEP | - | .9920(24) | - |
| FastICA | .9998 | .8327 | .9173 |
| Mixed | .5278 | .6174 | - |

task. This indicates that the overcompleteness of the task was in this case sufficiently helpful for making the problem separable up to some reasonable precision.

On the audio tasks (see Table 2), the results for the linear models were of very high precision but not better than those obtained with FastICA, unless one would be able to select settings based on the correlation scores directly. This still indicates that the method is able to learn the task at high precision without explicitly using properties which are specific to the linear problem. On the PNL task, the resampling based model scored better than the baseline (although we admit it may not be a very strong one). The Anica-g model scored worse when the hyper-parameter selection procedure was used but the score of the best working settings suggests that it might do similarly well as the resampling model with a better model selection procedure. See the appendix for some reconstruction plots of some of the individual models.

To get more insight in the convergence behavior of the models, we plotted the correlations with the source signals, the discriminator costs and the reconstruction costs of two linear models for the synthetic signals in Figure 4. For both a GAN and a WGAN version of the resampling-based model, the recognition and discriminator costs seem to be informative about the convergence of the correlation scores. However, we also observed situations in which the losses made a sudden jump after being stuck at a suboptimal value for quite a while and this might indicate why the consistency of the scores may be more important than their individual values.

Table 2: Maximum correlation results on all tasks for the audio data. A 'g' in the suffix of the model name indicates that a separate generator network was used instead of the resampling method. Two scores separated by a '/' indicate that the first score was obtained using the model selection described in Section 6.1 while the second score is simply the best score *a posteriori*. Parentheses refer contain the standard deviation of the scores multiplied with $10^{-4}$.

| Method | Linear | PNL |
|---|---|---|
| Anica | .9996(4.9)/1.0(.1) | .9929(18)/.9948(12) |
| Anica-g | .9996(3.1)/1.0(.1) | .9357(671)/.9923(19) |
| PNLMISEP | - | .9567(471) |
| FastICA | 1.0 | .8989 |
| Mixed | .5338 | .6550 |

## 7 DISCUSSION

As our results showed, adversarial objectives can successfully be used to learn independent features in the context of non-linear ICA source separation. We showed that the methods can be applied to a variety of architectures, work for signals that are both sub-gaussian and super-gaussian. The method were also able so separate recordings of human speech.

A serious difficulty with evaluating general methods for learning independent features is that the non-linear ICA problem is ill-posed. Ideally, we should use a general measure of both the independence and amount of information in the learned features. Both quantities are very hard to estimate in higher dimensions and preliminary attempts to use either the Hilbert Schmidt Independence Criterion (Gretton et al., 2005) and nearest-neighbor methods (Kraskov et al., 2004) for model selection haven't been succesful yet. We think that our overcomplete learning setup and heuristic model selection method are first steps in the way to evaluating these models but more principled approaches are desperately needed. We hope that future work on convergence measures for GANs will also improve the practical applicability of our methods by allowing for more principled model selection.

To conclude, our results show that adversarial objectives can be used to maximize independence and solve linear and non-linear ICA problems without relying on specific properties of the mixing process. While the ICA models we implemented are not always easy to optimize, they seem to work well in practice and can easily be applied to various different types of architectures and problems. Future work should be devoted to a more thorough theoretical analysis of of the proposed methods for minimizing and measuring dependence and how to evaluate them.

ACKNOWLEDGMENTS

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

## A   SYNTHETIC SIGNALS

The synthetic signals were defined as follows:

$$s_1(t) = \text{sign}(\cos(310\pi t)),$$
$$s_2(t) = \sin(1600\pi t),$$
$$s_3(t) = \sin(600\pi t + 6\cos(120\pi t)),$$
$$s_4(t) = \sin(180\pi t),$$
$$s_5(t) \sim \text{uniform}(x|[-1, 1]),$$
$$s_6(t) \sim \text{laplace}(x|\mu = 0, b = 1).$$

The experiments were done using the first 4000 samples with $t$ linearly spaced between $[0, 0.4]$.

## B   FIGURES

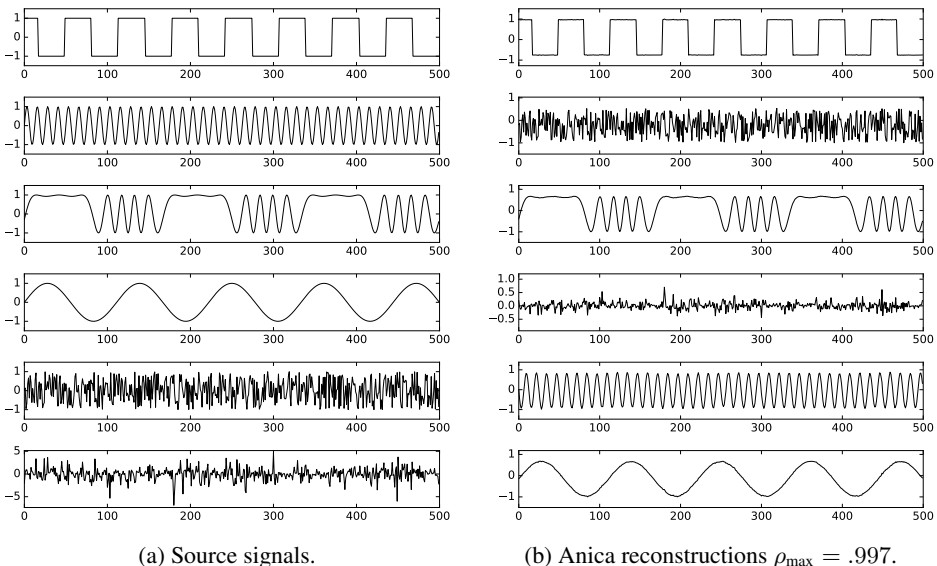

(a) Source signals.                    (b) Anica reconstructions $\rho_{\max} = .997$.

Figure 5: Sources and reconstructions for the linear synthetic source ICA task. The predictions have been rescaled to lie within the range $[-1, 1]$ for easier comparison with the source signals. This causes the laplacian samples to appear scaled down. The scores $\rho_{\max}$ represent the maximum absolute correlation over all possible permutations of the signals.

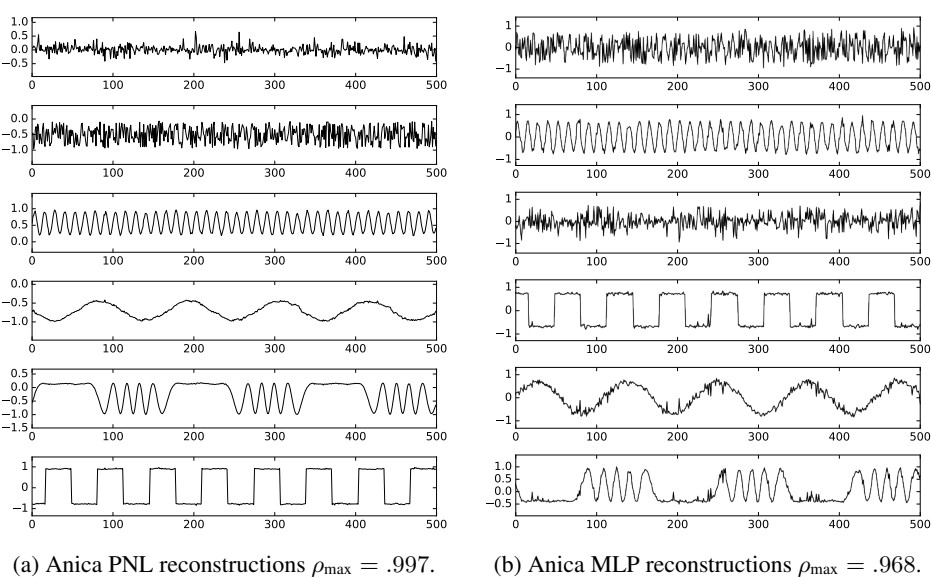

(a) Anica PNL reconstructions $\rho_{\max} = .997$.    (b) Anica MLP reconstructions $\rho_{\max} = .968$.

Figure 6: Reconstructions for the post-nonlinear mixture and MLP mixture of the synthetic sources.

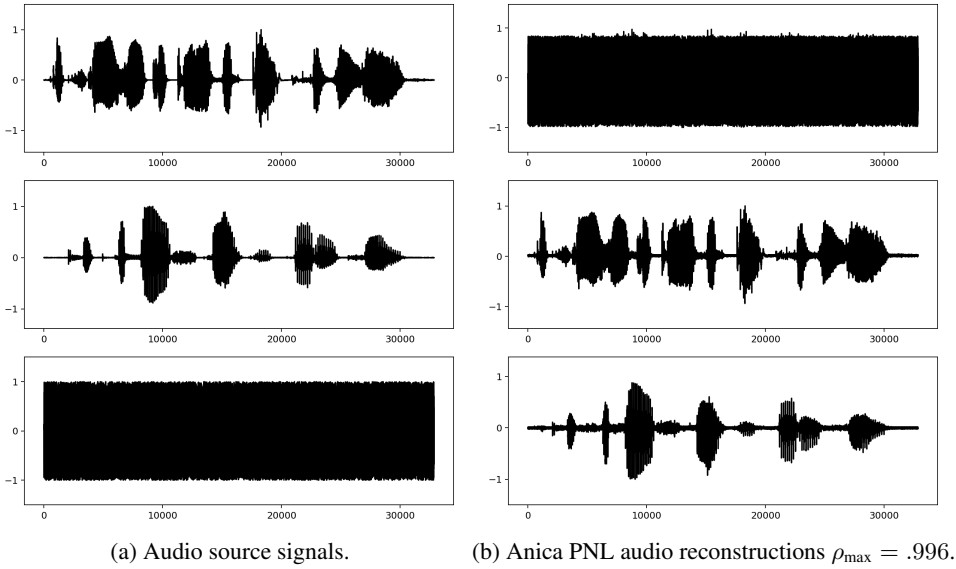

(a) Audio source signals.    (b) Anica PNL audio reconstructions $\rho_{\max} = .996$.

Figure 7: Sources and reconstructions for the post-nonlinear mixture of audio signals.

