# OpenReview forum: "Learning Independent Features with Adversarial Nets for Non-linear ICA"
_ICLR.cc/2018/Conference — Reject_

### Official Review · AnonReviewer3 · 2017-11-25
**Proposed Wasserstein GAN: not well-suited to ICA**

**Rating:** 3
**Confidence:** 5

**Review:**

The focus of the paper is independent component analysis (ICA) and its nonlinear variants such as the post non-linear (PNL) ICA model. Motivated by the fact that estimating mutual information and similar dependency measures require density estimates and hard to optimize, the authors propose a Wasserstein GAN (generative adversarial network) based solution to tackle the problem, with illustrations on 6 (synthetic) and 3-dimemensional (audio) examples. The primary idea of the paper is to use the Wasserstein distance as an independence measure of the estimated source coordinates, and optimize it in a neural network (NN) framework.

Although finding novel GAN applications is an exciting topic, I am not really convinced that ICA with the proposed Wasserstein GAN based technique fulfills this goal.

Below I detail my reasons:

1)The ICA problem can be formulated as the minimization of pairwise mutual information [1] or one-dimensional entropy [2]. In other words, estimating the joint dependence of the source coordinates is not necessary; it is worthwhile to avoid it.

2)The PNL ICA task can be efficiently tackled by first 'removing' the nonlinearity followed by classical linear ICA; see for example [3].

3)Estimating information theoretic (IT) measures (mutual information, divergence) is a quite mature field with off-the-self techniques, see for example [4,5,6,8]. These methods do not estimate the underlying densities; it would be superfluous (and hard).

4)Optimizing non-differentiable IT measures can computationally quite efficiently carried out in the ICA context by e.g., Givens rotations [7]; differentiable ICA cost functions can be robustly handled by Stiefel manifold methods; see for example [8,9].

5)Section 3.1: This section is devoted to generating samples from the product of the marginals, even using separate generator networks. I do not see the necessity of these solutions; the subtask can be solved by independently shuffling all the coordinates of the sample.

6)Experiments (Section 6):
i) It seems to me that the proposed NN-based technique has some quite serious divergence issues: 'After discarding diverged models, ...' or 'Unfortunately, the model selection procedure also didn't identify good settings for the Anica-g model...'.
ii) The proposed method gives pretty comparable results to the chosen baselines (fastICA, PNLMISEP) on the selected small-dimensional tasks. In fact, [7,8,9] are likely to provide more accurate (fastICA is a simple kurtosis based method, which is
a somewhat crude 'estimate' of entropy) and faster estimates; see also 2).

References:
[1] Pierre Comon. Independent component analysis, a new concept? Signal Processing, 36:287-314, 1994.
[2] Aapo Hyvarinen and Erkki Oja. Independent Component Analysis: Algorithms and Applications. Neural Networks, 13(4-5):411-30, 2000.
[3] Andreas Ziehe, Motoaki Kawanabe, Stefan Harmeling, and Klaus-Robert Muller. Blind separation of postnonlinear mixtures using linearizing transformations and temporal decorrelation. Journal of Machine Learning Research, 4:1319-1338, 2003.
[4] Barnabas Poczos, Liang Xiong, and Jeff Schneider. Nonparametric divergence: Estimation with applications to machine learning on distributions. In Conference on Uncertainty in Artificial Intelligence, pages 599-608, 2011.
[5] Arthur Gretton, Karsten M. Borgwardt, Malte J. Rasch, Bernhard Scholkopf, Alexander Smola. A Kernel Two-Sample Test. Journal of Machine Learning Research, 13:723-773, 2012.
[6] Alan Wisler, Visar Berisha, Andreas Spanias, Alfred O. Hero. A data-driven basis for direct estimation of functionals of distributions. TR, 2017. (https://arxiv.org/abs/1702.06516)
[7] Erik G. Learned-Miller, John W. Fisher III. ICA using spacings estimates of entropy. Journal of Machine Learning Research, 4:1271-1295, 2003.
[8] Francis R. Bach. Michael I. Jordan. Kernel Independent Component Analysis. Journal of Machine Learning Research 3: 1-48, 2002.
[9] Hao Shen, Stefanie Jegelka and Arthur Gretton. Fast Kernel-Based Independent Component Analysis, IEEE Transactions on Signal Processing, 57:3498-3511, 2009.

---

> ### Author Response · Authors · 2018-01-05
> **Thanks for the feedback.**
>
> Thanks for the feedback and interesting references.
>
> Many of the criticisms here seem to be based on notions which are specific to linear ICA. Unfortunately this seems to be attributable to a lack of clarity in the paper and we'd like to emphasize that we didn't try to provide an alternative to methods which have been specifically designed for that problem. We evaluated our methods on linear ICA and PNL ICA because solutions to these problems are known and comparisons were possible but the point is that the method we propose is less dependent on the specific mixing process.
>
> "1)The ICA problem can be formulated as the minimization of pairwise mutual information [1] or one-dimensional entropy [2]. In other words, estimating the joint dependence of the source coordinates is not necessary; it is worthwhile to avoid it."
>
> The first observation is specific to the linear case but interesting to know about. Working with the entropy seems to be based on the same ideas as infomax and introduces other limitations but we consider it complementary to our approach.
>
> "2)The PNL ICA task can be efficiently tackled by first 'removing' the nonlinearity followed by classical linear ICA; see for example [3]."
>
> While we didn't aim to be optimal for the PNL case either, we'd like to point out that the approach in [3] is still an iterative procedure.
>
> "4)Optimizing non-differentiable IT measures can computationally quite efficiently carried out in the ICA context by e.g., Givens rotations [7]; differentiable ICA cost functions can be robustly handled by Stiefel manifold methods; see for example [8,9]."
>
> These points seem to be specific to the linear case again but are once again interesting.
>
> "5)Section 3.1: This section is devoted to generating samples from the product of the marginals, even using separate generator networks. I do not see the necessity of these solutions; the subtask can be solved by independently shuffling all the coordinates of the sample."
>
> The first solution is indeed basically shuffling the coordinates of the sample but we admit that the text was a bit overly didactic and we shortened it a bit. The separate generator networks could be interesting in a setup in which shuffling is not desirable because there are temporal dependencies, for example. We changed the text to make this more clear.
>
> "6)Experiments (Section 6):
> i) It seems to me that the proposed NN-based technique has some quite serious divergence issues: 'After discarding diverged models, ...' or 'Unfortunately, the model selection procedure also didn't identify good settings for the Anica-g model...'.
> ii) The proposed method gives pretty comparable results to the chosen baselines (fastICA, PNLMISEP) on the selected small-dimensional tasks. In fact, [7,8,9] are likely to provide more accurate (fastICA is a simple kurtosis based method, which is a somewhat crude 'estimate' of entropy) and faster estimates; see also 2)."
>
> The first point is fair in that our model selection heuristic wasn't always able to identify the best model and that GAN training can be unstable. That said, the discarding of models was mainly because we performed a random search with aggressive hyperparameter ranges which could select very high learning rates, for example. The second point is fair too in that the cited methods might prove to be stronger baselines. We don't think that obtaining comparable results with a more general method is a bad thing but that is of course somewhat subjective.
>
> We'd finally like to point out that we don't propose the use of the Wasserstein GAN loss specifically but GAN type objectives in general for learning independent features. The WGAN example in the text was mainly there to illustrate how in some cases the objective can be seen as a proxy for the mutual information.
>
> Thanks again.

---

> > ### Comment · AnonReviewer3 · 2018-01-12
> > **Thank you for the feedback**
> >
> > Thank you for your response.
> >
> > Maximizing independence under general mixing conditions does not necessarily lead to the recovery of the underlying independent sources (even up to the standard ambiguities); this is one of the major motivations why the linear and post-nonlinear ICA (PNL-ICA) tasks have been considered in the literature.
> >
> > Constructing new general ICA 'solvers' can have certain impact, however the merits of the proposed heuristic are not illustrated/clear.
> > 1)In case of linear and post-nonlinear ICA: Available off-the-shelf methods can solve 1-2 orders-of-magnitude larger tasks than the ones studied with high accuracy in a numerically robust way.
> > 2)For general non-linear ICA tasks:
> > -One should investigate whether techniques maximizing an independence measure lead to provable solution, find the hidden sources.
> > -In fact, using approximate independence measures [such as (4)] raises further unhandled issues.
> >
> > To sum up, it would be crucial to (i) understand the validity domain of the studied scheme, (ii) make it comparable to existing methods (in terms of scalability, precision and robustness; at least in the ICA and PNL-ICA settings), and (iii) construct new well-posed non-linear ICA tasks.
> >
> > My opinion has not changed.

---

### Official Review · AnonReviewer1 · 2017-11-26
**Interesting nonlinear ICA method, but unfocused presentation and poor comparisons**

**Rating:** 5
**Confidence:** 5

**Review:**

The paper proposes a GAN variant for solving the nonlinear independent component analysis (ICA) problem. The method seems interesting, but the presentation has a severe lack of focus.

First, the authors should focus their discussion instead of trying to address a broad range of ICA problems from linear to post-nonlinear (PNL) to nonlinear. I would highly recommend the authors to study the review "Advances in Nonlinear Blind Source Separation" by Jutten and Karhunen (2003/2004) to understand the problems they are trying to solve.

Linear ICA is a solved problem and the authors do not seem to be able to add anything there, so I would recommend dropping that to save space for the more interesting material.

PNL ICA is solvable and there are a number of algorithms proposed for it, some cited already in the above review, but also more recent ones. From this perspective, the presented comparison seems quite inadequate.

Fully general nonlinear ICA is ill-posed, as shown already by Darmois (1953, doi:10.2307/1401511). Given this, the authors should indicate more clearly what is their method expected to do. There are an infinite number of nonlinear ICA solutions - which one is the proposed method going to return and why is that relevant? There are fewer relevant comparisons here, but at least Lappalainen and Honkela (2000) seem to target the same problem as the proposed method.

The use of 6 dimensional example in the experiments is a very good start, as higher dimensions are quite different and much more interesting than very commonly used 2D examples.

One idea for evaluation: comparison with ground truth makes sense for PNL, but not so much for general nonlinear because of unidentifiability. For general nonlinear ICA you could consider evaluating the quality of the estimated low-dimensional data manifold or evaluating the mutual information of separated sources on new test data.

Update after author feedback: thanks for the response and the revision. The revision seems more cosmetic and does not address the most significant issues so I do not see a need to change my evaluation.

---

> ### Author Response · Authors · 2018-01-05
> **Thanks for the feedback.**
>
> We first like to thank the reviewer for the valuable feedback and suggestions.
>
> We acknowledge that the linear and PNL ICA problems are more or less solved. However, we respectfully disagree that we should drop the treatment of these problems because we still think it is interesting that they can be solved with a new approach which in our opinion is very different from previous methods. This was not obvious to us when we started our research.
>
> A better definition of the version of the non-linear problem would indeed have been desirable in the context of source separation. While we presented the overcomplete case as a first step for evaluating the method, we surely realize that it doesn't come with any theoretical guarantees and that the obtained correlation scores are limited in interpretability. We adjusted the text to make this more clear.
>
> The alternative to use estimates of the mutual information is certainly something we considered for both evaluation and model selection but this proved to be difficult in general. We tried both Kraskov's nearest-neighbor estimator and the Hilbert Schmidt Independence Criterion but both these estimators typically seemed to consider the features fully independent during most stages of training and don't take into account how informative they are about the input. We still like to thank the reviewer for motivating us to pursue this direction further and like to hear more in detail what is meant with the "quality of the low-dimensional data manifold". If there is some principled way of measuring the latter we would certainly like to investigate it.
>
> Thanks.

---

### Official Review · AnonReviewer2 · 2017-11-26
**Thought provoking paper but lacks more detailed analysis**

**Rating:** 6
**Confidence:** 3

**Review:**


The idea of ICA is constructing a mapping from dependent inputs to outputs (=the derived features) such that the outputs are as independent as possible. As the input/output densities are often not known and/or are intractable, natural independence measures such as mutual information are hard to estimate. In practice, the independence is characterized by certain functions of higher order moments -- leading to several alternatives in a zoo of independence objectives.

The current paper makes the iteresting observation that independent features can also be computed via adversarial objectives. The key idea of adversarial training is adapted in this context as comparing samples from the joint distribution and the product of the marginals.

Two methods are proposed for drawing samples from the products of marginals.
One method is generating samples but permuting randomly the sample indices for individual marginals - this resampling mechanism generates approximately independent samples from the product distribution. The second method is essentially samples each marginal separately.

The approach is demonstrated in the solution of both linear and non-linear ICA problems.

Positive:
The paper is well written and easy to follow on a higher level. GAN's provide a fresh look at nonlinear ICA and the paper is certainly thought provoking.


Negative:
Most of the space is devoted for reviewing related work and motivations, while the specifics of the method are described relatively short in section 4. There is no analysis and the paper is
somewhat anecdotal. The simulation results section is limited in scope. The sampling from product distribution method is somewhat obvious.


Questions:

- The overcomplete audio source separation case is well known for audio and I could not understand why a convincing baseline can not be found. Is this due to nonlinear mixing?
As 26 channels and 6 channels are given, a simple regularization based method can be easily developed to provide a baseline performance,


- The need for normalization in section 4 is surprising, as it obviously renders the outputs dependent.

- Figure 1 may be misleading as h are not defined

---

> ### Author Response · Authors · 2018-01-05
> **Thanks for the suggestions and comments. We adjusted the paper.**
>
> First of all, thanks for the feedback and suggestions.
>
> We removed some of the text which basically reiterated the sampling process as it seemed that multiple reviewers found it redundant. As you suggested, we made the definitions of the full system in Section 4 a bit more explicit.
>
> "The overcomplete audio source separation case is well known for audio and I could not understand why a convincing baseline can not be found. Is this due to nonlinear mixing?"
>
> Good question and something we certainly looked at. The most important reason for our lack of a baseline here is that the real-world audio separation setting is slightly more complicated due to the different arrival times of the source signals and reverberation. Most multi-channel audio separation methods we know of work in the frequency domain to alleviate some of these issues, introducing the necessity to predict phase information to reconstruct the raw signals. Another issue was that the evaluation criteria and benchmark data sets in this domain still seem to be under active development but of course we'd love to hear about good real-world benchmarks we might have overlooked.
>
> "As 26 channels and 6 channels are given, a simple regularization based method can be easily developed to provide a baseline performance, "
>
> This sounds interesting. Do you mean an auto-encoder with more standard regularization? In that case the hidden units wouldn't be trained to become independent but perhaps we misunderstood the suggestion.
>
> "The need for normalization in section 4 is surprising, as it obviously renders the outputs dependent."
>
> We normalize over samples in a batch and not over the units in the layer but admit that this was not clear in the paper. We changed the text to address this issue.
>
> We also removed the figure you referred to as it didn't add much and took up a lot of space.
>
> Thanks again.

---

### Decision · Program_Chairs · 2018-01-29
**ICLR 2018 Conference Acceptance Decision**

**Decision:**

Reject

**Comment:**

The paper proposes the use of GANs to match the joint distribution of features to the product of their marginals for ICA. The approach is totally plausible but reviewers have complaints about lack of rigor and analysis in terms of (i) mixing conditions under which the proposed GAN based approach will work, given that ICA is ill-posed for general nonlinear mixing  (ii) comparison with prior work on linear and PNL ICA.

Further, in most scenarios where GANs are used, one of the distributions is fixed (say, the real distribution) and the other is dynamic (fake distribution) trying to come close to the fixed distribution during optimization. In the proposed method, the discriminator encodes the distance b/w joint and product of marginals which are both dynamic during the learning. It might be useful to comment whether or not it has any implications wrt increased instability of training, etc.